# Significant sparse polygenic risk scores across 813 traits in UK Biobank

Yosuke Tanigawa[1,2]*, Junyang Qian[3], Guhan Venkataraman[1], Johanne Marie Justesen[1], Ruilin Li[4], Robert Tibshirani[1,3], Trevor Hastie[1,3], Manuel A. Rivas[1]*

**1** Department of Biomedical Data Science, Stanford University, Stanford, California, United States of America, **2** Computer Science and Artificial Intelligence Laboratory, Massachusetts Institute of Technology, Cambridge, Massachusetts, United States of America, **3** Department of Statistics, Stanford University, Stanford, California, United States of America, **4** Institute for Computational and Mathematical Engineering, Stanford University, Stanford, California, United States of America

\* tanigawa@mit.edu (YT); mrivas@stanford.edu (MAR)

## Abstract

We present a systematic assessment of polygenic risk score (PRS) prediction across more than 1,500 traits using genetic and phenotype data in the UK Biobank. We report 813 sparse PRS models with significant ($p < 2.5 \times 10^{-5}$) incremental predictive performance when compared against the covariate-only model that considers age, sex, types of genotyping arrays, and the principal component loadings of genotypes. We report a significant correlation between the number of genetic variants selected in the sparse PRS model and the incremental predictive performance (Spearman's $\rho = 0.61$, $p = 2.2 \times 10^{-59}$ for quantitative traits, $\rho = 0.21$, $p = 9.6 \times 10^{-4}$ for binary traits). The sparse PRS model trained on European individuals showed limited transferability when evaluated on non-European individuals in the UK Biobank. We provide the PRS model weights on the Global Biobank Engine (https://biobankengine.stanford.edu/prs).

**Data Availability Statement:** The sparse PRS model weights generated from this study are available on the Global Biobank Engine (https://biobankengine.stanford.edu/prs). The significant PRS models are also available at the PGS catalog

## Author summary

Polygenic risk score (PRS), an approach to estimate genetic predisposition on disease liability by aggregating the effects across multiple genetic variants, has attracted increasing research interest. While there have been improvements in the predictive performance of PRS for some traits, the applicability of PRS models across a wide range of human traits has not been clear. Here, applying penalized regression using Batch Screening Iterative Lasso (BASIL) algorithm to more than 269,000 individuals of white British ancestry in UK Biobank, we systematically characterize PRS models across more than 1,500 traits. We report 813 traits with PRS models of statistically significant predictive performance. While the statistical significance does not necessarily directly translate into clinical relevance, we investigate the properties of the 813 significant PRS models and report a significant correlation between predictive performance and estimated SNP-based heritability. We find that the number of genetic variants selected in our sparse PRS model is significantly correlated with the incremental predictive performance in both quantitative and binary traits.

(https://www.pgscatalog.org/publication/PGP000244/ and https://www.pgscatalog.org/publication/PGP000128/, score IDs are listed in S1 Table). The BASIL algorithm implemented in the R *snpnet* package was used in the PRS analysis, which is available at https://github.com/rivas-lab/snpnet. The analyses presented in this study were based on the individual-level data accessed through the UK Biobank: https://www.ukbiobank.ac.uk.

**Funding:** This work has been supported by The National Human Genome Research Institute (NHGRI) of the National Institutes of Health (NIH) [R01HG010140 to M.A.R.]; NIH [5U01 HG009080 to M.A.R., 5R01 EB 001988-21 to T.H., and 5R01 EB001988-16 to R.T]; National Science Foundation [DMS-1407548 to T.H., 19 DMS1208164 to R.T.]; Stanford University School of Medicine [to Y.T., R. L., and M.A.R.]; and the Funai Foundation for Information Technology [to Y.T.]. The authors of this manuscript have received the following salary support: NHGRI of NIH [R01HG010140 to Y.T. and M.A.R., R01HG008155 to Y.T.]; NIH [5U01 HG009080 to M.A.R.]; and the National Institute on Aging of NIH [R01AG067151 to Y.T.]. The content is solely the responsibility of the authors and does not necessarily represent the official views of the funding agencies; funders had no role in study design, data collection and analysis, decision to publish, or preparation of the manuscript.

**Competing interests:** The corresponding authors have read the journal's policy and the authors of this manuscript have the following competing interests: M.A.R is a consultant at MazeTx and is currently on leave at HiBio.

Our transferability assessment of PRS models in UK Biobank revealed that the sparse PRS models trained on individuals of European ancestry had a lower predictive performance for individuals of African and Asian ancestry groups.

## Introduction

Polygenic risk score (PRS), an estimate of an individual's genetic liability to a trait or disease, has been proposed for disease risk prediction with potential clinical relevance for some traits [1,2]. Due to training data sample size increase and methods development advances for variable selection and effect size estimation, PRS predictive performance has improved [3–17]. However, it has not been clear what would be the predictive performance of PRS models when it is applied to a wide range of traits and their transferability across ancestry groups. Rich phenotypic information in large-scale genotyped cohorts provides an opportunity to address this question.

Here, we present significant sparse PRSs across 813 traits in the UK Biobank [18,19]. We applied the recently developed batch screening iterative lasso (BASIL) algorithm implemented in the R *snpnet* package [10] across more than 1,500 traits consisting of binary outcomes and quantitative traits, including disease outcomes and biomarkers, respectively (Fig 1, S1 Table). As opposed to most of the recently developed PRS methods that take genome-wide association study (GWAS) summary statistics as input, BASIL/*snpnet* is capable of performing variable selection and effect size estimation simultaneously from individual-level genotype and phenotype data. BASIL/*snpnet* results in sparse PRS models, meaning that most genetic variants in the input dataset have zero coefficient. For example, the *snpnet* PRS for standing height, a classic example of polygenic traits, includes 51,209 variants, which has non-zero coefficients for 4.7% of 1,080,968 genetic variants and allelotypes present in the input genetic data. Moreover, this approach does not require the explicit specification of the underlying genetic architecture of traits, suitable for a phenome-wide application of PRS modeling. Using individuals in a hold-out test set, we evaluated their predictive performance and their statistical significance, resulting in 813 significant ($p < 2.5 \times 10^{-5}$) PRS models. We find a significant correlation between the number of the genetic variants selected in the model and the incremental predictive performance compared to the covariate-only models across quantitative traits and binary traits. We assess the transferability of the PRS models across ancestry groups using individuals from non-British white, African, South Asian, and East Asian ancestry in the UK Biobank. We make the coefficients of the PRS models publicly available via the PRS map web application on the Global Biobank Engine [20] (https://biobankengine.stanford.edu/prs).

## Results

### Characterizing sparse PRS models with BASIL algorithm

To build sparse PRSs across a wide range of phenotypes, we compiled a total of 1,565 traits in the UK Biobank. We grouped them into trait categories, such as disease outcomes, anthropometry measures, and cancer phenotypes (S1 Table, Methods). We analyzed a total of 1,080,968 genetic variants and allelotypes from the directly-genotyped variants [19], imputed HLA allelotypes [21], and copy number variants [22]. Using 80% (n = 269,704) of unrelated individuals of white British ancestry, we applied batch screening iterative lasso (BASIL) implemented in the R *snpnet* package [10]. This recently developed method characterizes PRS models by simultaneously performing variable selection and effect size estimation. Applying different levels of penalization in the Lasso regression with penalty factors, we prioritized the medically relevant

## (A) Study cohort and phenotypes

377,066 unrelated individuals in UK Biobank (**S2 Table**)
337,129 individuals of white British ancestry
- score development (n = 269,704)
- hold-out test set (n = 67,425)
Additional hold-out sets for transferability assessment
- non-British white (n = 24,905)
- African (n = 6,497)
- South Asian (n = 7,831)
- East Asian (n = 1,704)

1,565 phenotypes in UK Biobank (**S1 Table**)
871 quantitative traits
Anthropometry, biomarkers, blood assays, bone densitometry ...
694 binary traits
Disease outcomes, cancer, lifestyle factors, family history ...

## (B) Sparse PRS models across UK Biobank traits

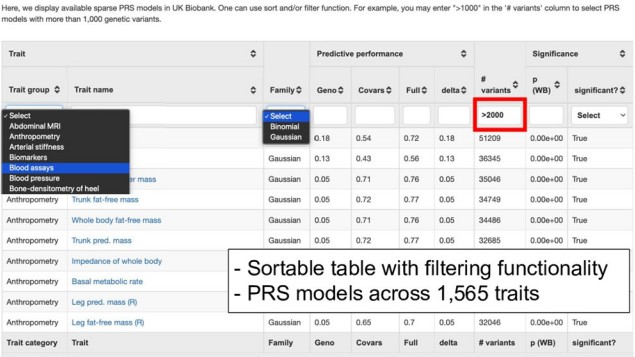

## (C) Prediction and PRS weights for quantitative traits

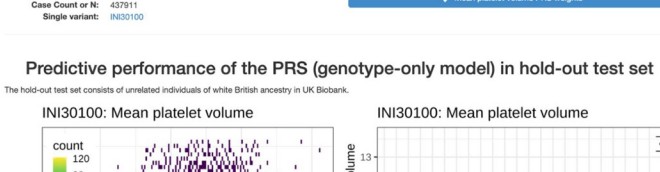

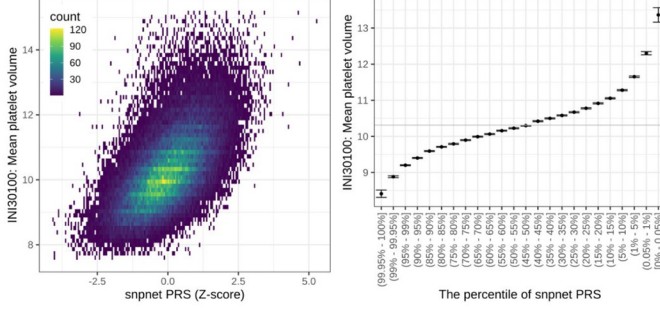

## (D) Prediction and PRS weights for binary traits

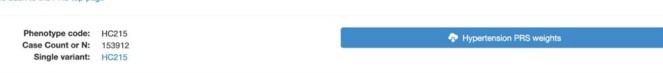

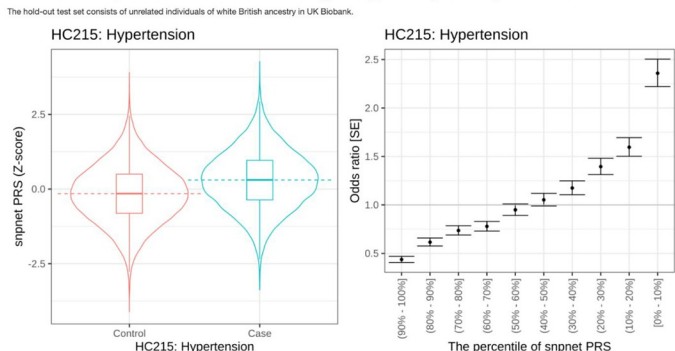

## (E) Non-zero coefficients of sparse PRS model

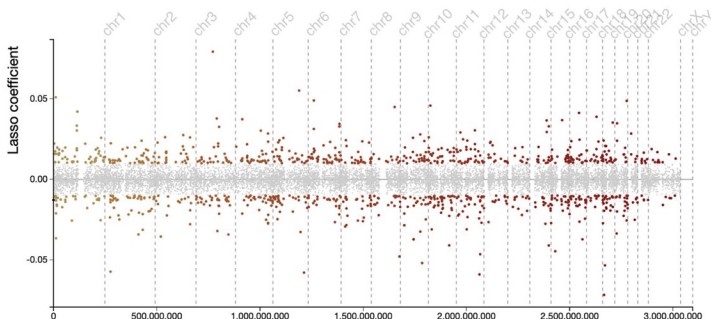

Note: only 13791 variants in active path for Lasso are included in the coefficient plot.

## (F) Transferability assessment in UK Biobank

| Metric | Ancestry group | Predictive performance | | | | n | case_n | control_n |
|---|---|---|---|---|---|---|---|---|
| | | Geno | Covars | Full | delta | | | |
| Nagelkerke's pseudo-$R^2$ | white British (model development) | 0.240 | 0.113 | 0.322 | 0.209 | 269704 | 91884 | 177820 |
| Nagelkerke's pseudo-$R^2$ | white British (hold-out test set) | 0.065 | 0.113 | 0.179 | 0.066 | 67425 | 22882 | 44543 |
| Nagelkerke's pseudo-$R^2$ | Non-British white | 0.055 | 0.138 | 0.191 | 0.054 | 24905 | 7648 | 17257 |
| Nagelkerke's pseudo-$R^2$ | South Asian | 0.038 | 0.175 | 0.213 | 0.038 | 7831 | 3161 | 4670 |
| Nagelkerke's pseudo-$R^2$ | East Asian | 0.055 | 0.177 | 0.226 | 0.048 | 1704 | 446 | 1258 |
| Nagelkerke's pseudo-$R^2$ | African | 0.011 | 0.175 | 0.177 | 0.003 | 6497 | 2919 | 3578 |
| Tjur's pseudo-$R^2$ | white British (model development) | 0.181 | 0.082 | 0.246 | 0.165 | 269704 | 91884 | 177820 |
| Tjur's pseudo-$R^2$ | white British (hold-out test set) | 0.047 | 0.082 | 0.132 | 0.050 | 67425 | 22882 | 44543 |
| Tjur's pseudo-$R^2$ | Non-British white | 0.040 | 0.100 | 0.141 | 0.041 | 24905 | 7648 | 17257 |
| Tjur's pseudo-$R^2$ | South Asian | 0.029 | 0.134 | 0.163 | 0.029 | 7831 | 3161 | 4670 |
| Tjur's pseudo-$R^2$ | East Asian | 0.039 | 0.130 | 0.168 | 0.039 | 1704 | 446 | 1258 |
| Tjur's pseudo-$R^2$ | African | 0.008 | 0.135 | 0.136 | 0.001 | 6497 | 2919 | 3578 |

**Fig 1. Significant sparse polygenic risk scores (PRSs) across 813 traits in the UK Biobank.** (A) We analyzed a total of more than 378,000 unrelated individuals and 1,565 traits in UK Biobank. We used 80% of individuals of white British ancestry for score development. For evaluation, we used the remaining 20% of individuals and additional individuals in other ancestry groups. (B) The full list of 1,565 traits with predictive performance is shown as a sortable table at Global Biobank Engine (https://biobankengine.stanford.edu/prs). (C) The predictive performance of PRS models for quantitative traits is summarized as a heatmap comparing the predicted risk score (Z-score) and observed trait value (left) and mean and standard error of trait values stratified by percentile bin (right). (D) The predictive performance of PRS models for binary traits is summarized as PRS score distribution stratified by case/control status (left) and odds ratio stratified by percentile bin (right). (E) The non-zero coefficients of the sparse PRS model are shown. (F) The predictive performance evaluation in training and test sets consist of individuals of white British ancestry, as well as additional sets consisting of individuals from non-British white, African, South Asian, and East Asian ancestry groups in the UK Biobank.

alleles in the PRS model. Specifically, we used the predicted consequence of the genotyped variants and the pathogenicity information in the ClinVar database. We prioritized protein-truncating variants, protein-altering variants, imputed HLA allelotype, and known pathogenic and likely-pathogenic variants by assigning lower penalty factors (Methods). As unpenalized covariates, we included age, sex, and the loadings of the top ten principal components (PCs) of genotypes. For 35 blood and urine biomarker traits, we took the *snpnet* PRS models from a recently published study [23], where the PRS models were characterized with the same methods on the same set of individuals following the adjustment for an extensive list of technical covariates, including fasting time and dilution factors, as well as for age, sex, and genotype PCs.

To evaluate the predictive performance ($R^2$ for quantitative traits and observed scale Nagelkerke's pseudo-$R^2$ [also known as Cragg and Uhler's pseudo-$R^2$] [24,25] for binary traits) and its statistical significance, we focused on the remaining 20% of unrelated individuals in the hold-out test set (n = 67,425) as well as additional sets of unrelated individuals in the following ancestry groups in UK Biobank: non-British European (non-British white, n = 24,905), African (n = 6,497), South Asian (n = 7,831), and East Asian (n = 1,704) (S2 Table, Methods). We found 813 PRS models with significant (p < 2.5 x $10^{-5}$ = 0.05/2,000, adjusted for multiple hypothesis testing with Bonferroni method) predictive performance in the hold-out test set of white British individuals (Methods). For the binary traits, we also evaluated the receiver operating characteristic area under the curve [ROC-AUC] and Tjur's Coefficient of Discrimination (Tjur's pseudo-$R^2$) [26].

The participants of the UK Biobank were genotyped on two different arrays: about 10% of participants were genotyped on the UK BiLEVE Axiom array, whereas the rest were genotyped on the UK Biobank Axiom array [19]. To account for the potential biases correlated with the types of arrays, we evaluated the predictive performance of the PRS by accounting for the types of the arrays in addition to the age, sex, and the top ten genotype PCs. We found the identity of the UK Biobank assessment centers mostly has a non-significant impact on the predictive performance (S1 Fig, Methods).

To assess the degree of prioritization of the medically relevant alleles, we selected standing height, body mass index (BMI), high cholesterol, and asthma. We compared the predictive performance and the number of genetic variants for each functional category. For the four selected traits, we found a little difference in the predictive performance ($R^2$ = 0.177 vs. 0.176 for the PRS model with penalty factor and without penalty factor, respectively, for standing height, $R^2$ = 0.111 vs. 0.111 for BMI, AUC = 0.620 vs. 0.619 for high cholesterol, and AUC = 0.617 vs. 0.617 for asthma) (S2 Fig) while we saw an enrichment of the number of the medically relevant alleles with non-zero coefficients in the PRS model with prioritization (2.14 fold enrichment standing height, 2.75 fold for BMI, 4.14 fold for high cholesterol, and 4.33 fold asthma) (Table 1 and S3 Table), highlighting the flexibility of the BASIL/*snpnet* in assigning different levels of penalization based on the variant-level information.

With the same set of four traits, we asked whether including the imputed genetic variants could improve the predictive performance. We saw some gain in the predictive performance in three traits but not for standing height (S3 Fig). Based on those results, we decided to move on to the phenome-wide application of the BASIL algorithm implemented in the R *snpnet* packages on the directly genotyped variants, imputed allelotypes, and copy number variants while prioritizing the medically relevant alleles with penalty factors.

## Significance and estimated effect size of sparse PRS models

We estimated the SNP-based heritability by applying linkage disequilibrium (LD) score regression (LDSC) [27] on genome-wide association study (GWAS) summary statistics. We

**Table 1. The prioritization of the medically relevant alleles with penalty factors.** The numbers of the genetic variants or allelotypes with non-zero coefficient values are shown for the selected four traits. The denominator represents the total number of variables included in the model. The numerator represents the number of the medically relevant alleles, which are one of the following: protein-truncating variants, protein-altering variants, imputed HLA allelotypes, the pathogenic or likely-pathogenic variants in the ClinVar database. The enrichment of the medically relevant variants is also shown.

| Trait | Number of selected genetic variants or allelotypes | | |
|---|---|---|---|
| | with penalty factor | without penalty factor | enrichment |
| Standing height | 4187 / 51209 | 2129 / 55937 | 2.15 |
| Body mass index | 2543 / 27126 | 977 / 28667 | 2.75 |
| High cholesterol | 969 / 5987 | 215 / 5506 | 4.14 |
| Asthma | 1022 / 6430 | 250 / 6819 | 4.34 |

compared it against the predictive performance ($R^2$ for quantitative traits and Nagelkerke's pseudo-$R^2$ for binary traits) of the significant PRS models (Fig 2). Across 244 binary traits and 569 quantitative traits with significant PRS models, we found higher estimated observed scale heritability for quantitative traits. Overall, we found a significant correlation between the estimated SNP-based observed scale heritability and predictive performance (Spearman's rank correlation coefficient ρ = 0.44, p-value = 3.5 x $10^{-13}$ for binary traits, ρ = 0.46, p-value = 1.4 x $10^{-31}$ for quantitative traits).

The basic covariates alone are already informative for phenotype prediction. To assess the incremental utility of PRSs, we quantified the incremental predictive performance by comparing the predictive performance of the full model that considers both genotypes and covariates and that of the covariate-only model across the 813 traits with significant sparse PRS. We found most traits have a modest increase in the effect sizes of the prediction with a few notable exceptions, such as celiac disease (Nagelkerke's pseudo-$R^2$ = 0.149 in the full model vs 0.006 in the covariate-only model, p = 3.8 x $10^{-162}$), hair color (red) (Nagelkerke's pseudo-$R^2$ = 0.603 vs. 0.008, p < 1 x $10^{-300}$), mean platelet volume ($R^2$ = 0.36 vs. 0.001, p < 1 x $10^{-300}$), heel bone mineral density ($R^2$ = 0.20 vs. 0.06, p < 1 x $10^{-300}$), and blood and urine biomarker traits [23] (Figs 3 and 4).

## Sparse PRS models offer an interpretation of genomic loci underlying the polygenic risk

Celiac disease is an autoimmune disorder that affects the small intestine from gluten consumption. The sparse PRS model for this trait, for example, consists of 428 variants that contain the imputed HLA allelotypes and variants near the MHC region in chromosome 6 [19,21]. The PRS model also contains genetic variants in all other autosomes, including a previously implicated missense variant in chromosome 12 (rs3184504, log(OR) = 0.15 in multivariate PRS model) in *SH2B3*. This gene encodes SH2B adaptor protein 3, which is involved in cellular signaling, hematopoiesis, and cytokine receptors [28] (Fig 4).

## The size of the PRS model is correlated with the incremental predictive performance

The significant PRS models have a wide range of the number of variables selected in the model, ranging from only one variable for iritis PRS (HLA allelotype, HLA-B*27:05, at the well-established HLA-B*27 locus [29,30]) to 51,209 variants selected for standing height PRS (Fig 5). We examined whether there is a relationship between the number of active variables in the significant PRS model and the incremental predictive performance. The significant

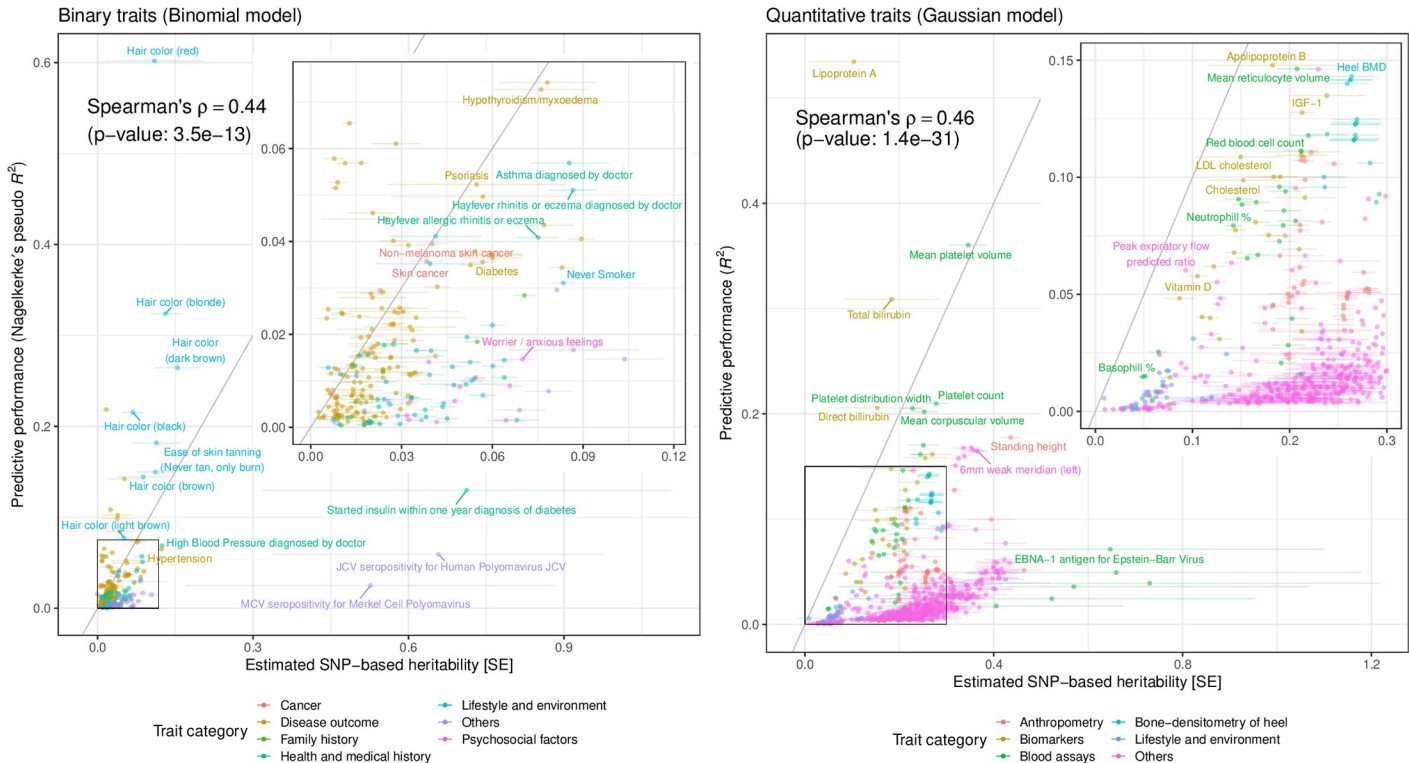

**Fig 2. Comparison of the estimated SNP-based heritability and predictive performance across the 813 traits with significant PRSs.** The predictive performance (Nagelkerke's pseudo-$R^2$ for 244 binary traits [left] and $R^2$ for 569 quantitative traits [right]) of the PRS models that only consider genetic variants are compared against the estimated SNP-based heritability. Both metrics are shown in observed scale and depend on the proportion of cases in the target and discovery cohorts. The solid gray lines represent y = x. We show the points on the bottom left corners in the inset plots. The error bars represent standard error. BMD: Bone mineral density.

correlation between the two quantities is stronger in quantitative (Spearman's rank correlation coefficient ρ = 0.61, p = 2.2 x $10^{-59}$) traits than in binary (ρ = 0.21, p = 9.6 x $10^{-4}$), reflecting the difference in power between binary and quantitative traits [31].

## Sparse PRS models exhibit limited transferability across ancestry groups

While the majority of the participants in the UK Biobank are of European ancestry, the inclusion of individuals from African and Asian ancestry enables an assessment of the transferability of the PRS models across ancestry groups in UK Biobank. In addition to the hold-out test set that we derived from the white British population, we focused on additional sets of individuals from non-British European (non-British white), African, South Asian, and East Asian ancestry groups and compared the incremental predictive performance with that in white British hold-out test set (Fig 6). For quantitative traits, the models predicted well for non-British white (linear regression fit of the incremental predictive performance: y = 0.91x), but they suffer limited transferability for the non-European ancestry groups (y = 0.56x, y = 0.47x, and y = 0.13x for South Asian, East Asian, and African, respectively). Similarly, in binary traits, the non-British white showed higher transferability (y = 0.80 x) than the non-European ancestry groups (y = 0.027x, y = 0.059x, and y = -0.145x for South Asian, East Asian, and African, respectively).

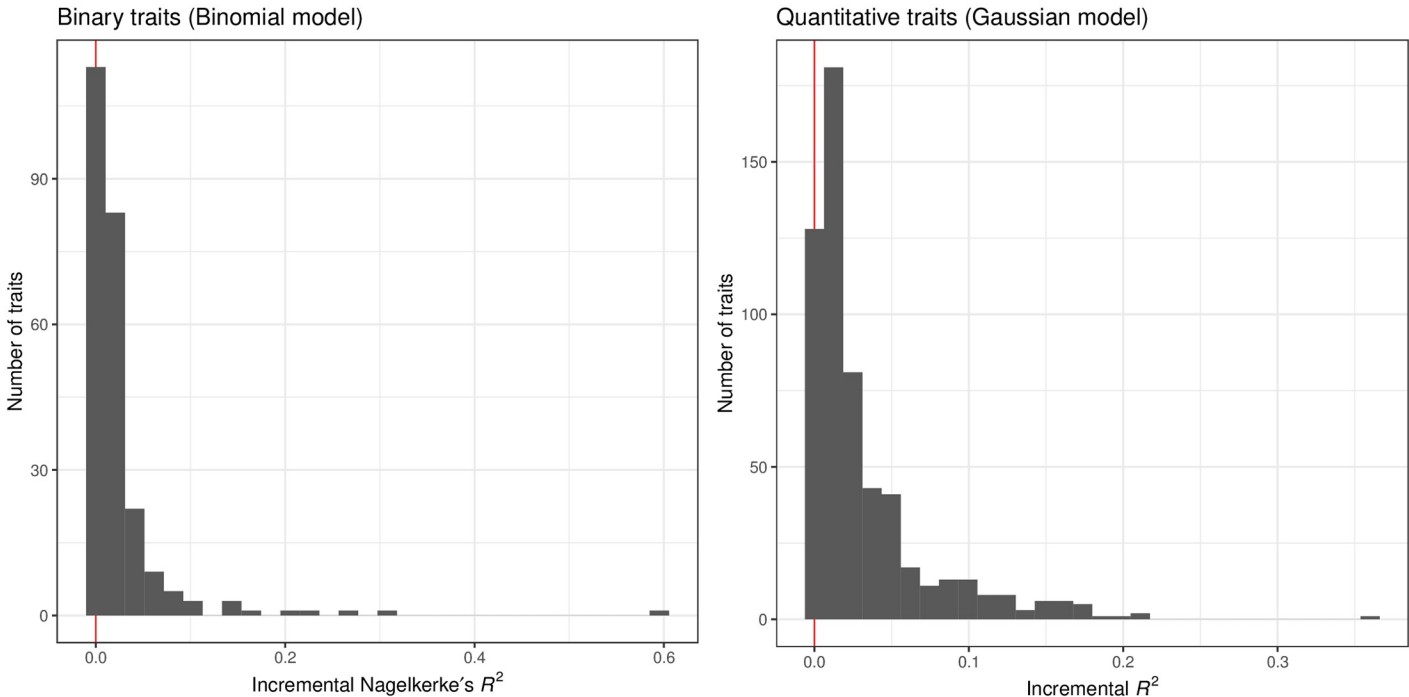

**Fig 3. Incremental predictive performance of PRS models across the 813 traits with significant predictive performance in the hold-out test set individuals of white British ancestry.** The predictive performance (Nagelkerke's pseudo-$R^2$ for 244 binary traits [left] and $R^2$ for 569 quantitative traits [right]) of the full models that consider both the genotype and covariates are compared against that of the covariate-only models, and their difference (the incremental predictive performance) are shown as a histogram.

## Discussion

In this study, we performed a systematic scan of polygenic prediction across more than 1,500 traits and reported 813 significant sparse PRS models. We found a correlation between the predictive performance of the significant PRS models and SNP-based heritability estimates. We assessed the effect size of the PRS model by quantifying the incremental predictive performance, which we define as the difference in the predictive performance between the covariate-only model and the full model consisting of both covariates and genetics. In both quantitative and binary traits, we find a significant correlation between the number of independent loci included in the model and their incremental predictive performance.

Our study is complementary to many other studies that focus on fewer traits to construct PRS models from GWAS meta-analysis and mixed models. While the sample size in our study is sufficiently large to observe statistical significance in predictive performance across hundreds of traits, it does not necessarily mean the clinical relevance of the PRS models. Moreover, population-based recruitment in UK Biobank may not be the best strategy to achieve the highest predictive performance for some traits. A disease-focused study [6,32–34] would be an attractive alternative strategy, especially when multiple genotyped cohorts recruited for the same disease are available or the disease of interest has a low population prevalence. Our study, instead, focused on the phenome-wide application of PRS across hundreds of traits in a single cohort by applying BASIL algorithm with readily available implementation in R *snpnet* package [10], which does not require explicit modeling of underlying genetic architecture across a wide variety of traits.

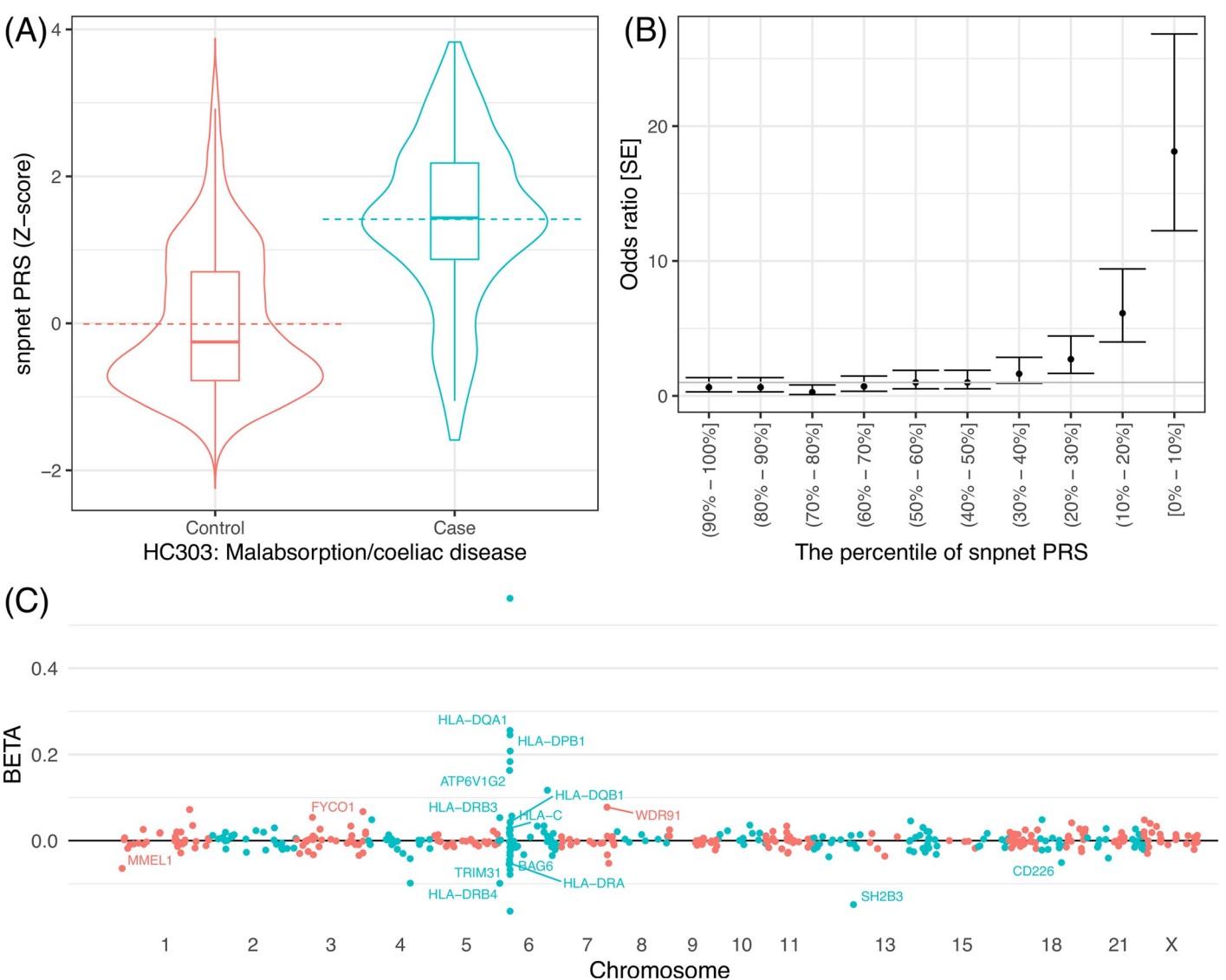

**Fig 4. The sparse PRS model and their predictive performance for celiac disease.** (A, B) the predictive performance of celiac disease PRS. (A) the celiac disease PRS distribution (y-axis) in a hold-out test set stratified by the disease case status (x-axis). The dashed lines represent the mean and the quantiles are shown as box plots. (B) The disease prevalence odds ratio compared to the individuals with middle (40–60 th percentile) PRS score stratified by PRS percentile bins. The error bars represent standard error (SE). (C) the coefficients of the celiac disease PRS model. The estimated effect size (y-axis) for each genetic variant (x-axis) is shown. The gene symbols are annotated in the plot for coding variants and HLA allelotypes with large effect size estimates.

For binary traits, we used observed scale pseudo-$R^2$ and observed scale SNP-based heritability estimates, given that population prevalence is available for only a subset of binary traits considered in the present study. Conversion to liability scale estimates will further enhance the validity of the comparison [35] and is of interest for future investigation.

Like other PRS approaches that consider datasets from one source population in the PRS training, our sparse model trained on the individual-level data of white British showed limited transferability across diverse ancestry groups [36–38]. The sample sizes of non-European ancestry groups in UK Biobank are smaller than that of European ancestry groups. In general, that will result in larger uncertainties in predictive performance assessment. Nonetheless,

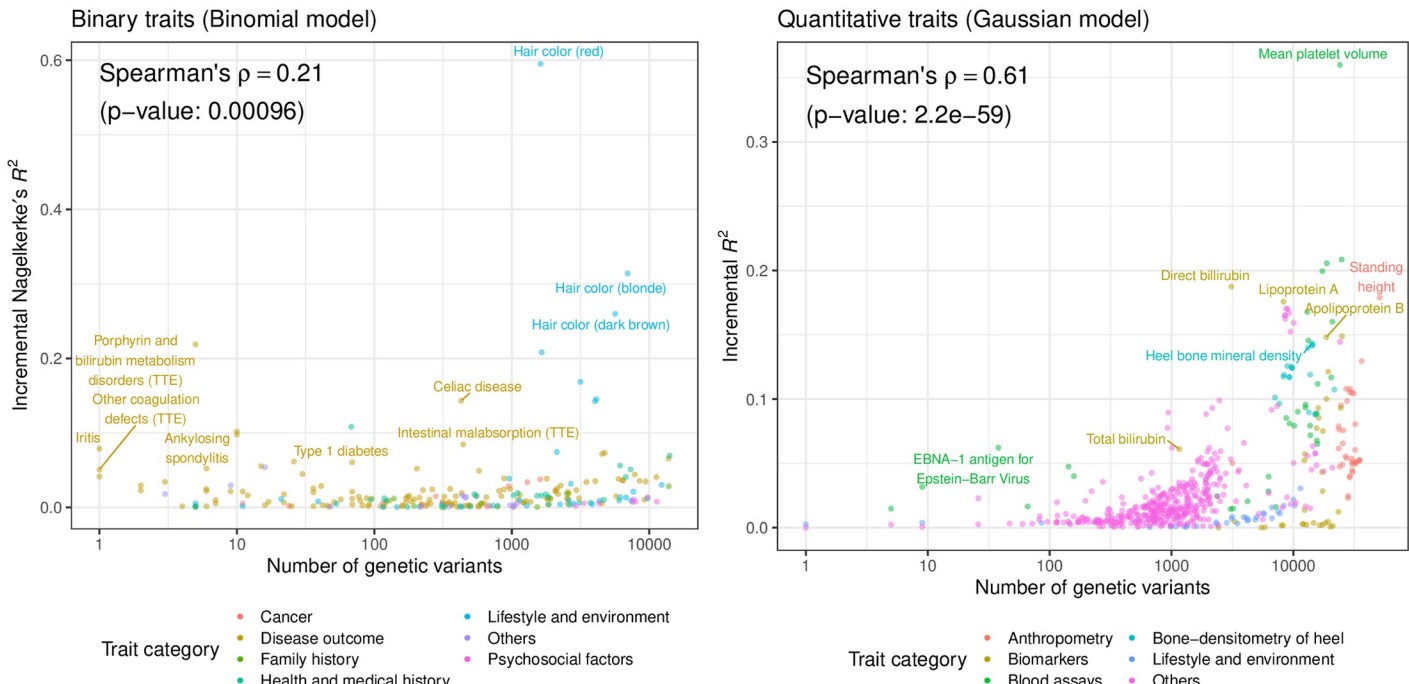

**Fig 5. Comparison of the effect size and the model size of sparse PRS.** The number of the genetic variants included in the model (size of the model, x-axis) and the incremental predictive performance (effect size of the model, y-axis) are shown for 244 binary traits (left) and 569 quantitative traits (right). TTE: time-to-event phenotype.

when we assess the incremental predictive performance across ancestry groups by comparing the full model consisting of the genetic data and basic covariates and the covariate-only model, we found the binary traits, including disease outcomes, have lower transferability compared to quantitative traits, including biomarkers, blood measurements, and anthropometric traits. The power difference between binary and quantitative traits [31], limitation in power for some traits, especially for the binary traits with limited case counts, and differences in heritability may be the contributing factors of the observed difference. Improvements of PRS models with high transferability across ancestry groups and the admixed individuals are of interest for future research.

Given the medical relevance [39–51], we prioritized pathogenic and likely-pathogenic variants reported in ClinVar [52] as well as predicted protein-truncating and protein-altering variants (Methods). Our analysis focusing on four traits suggests that prioritizing the medically relevant alleles does not necessarily improve the predictive performance. While our sparse PRS models show enrichment in the number of selected medically relevant alleles, there is no guarantee that the genetic variants included in the sparse PRS models were causal. It warrants further follow-up analysis with statistical fine-mapping and detailed functional characterization at each locus.

The increased availability of PRS models across multiple traits [17] exhibits a wide range of applications, including the improved genetic risk prediction of disease [23,53] and the identification of causal relationships across complex traits [54]. We provide the results on the Global Biobank Engine (https://biobankengine.stanford.edu/prs) as well as on the PGS catalog [17] and envision the resource will serve as an important basis to understand the polygenic basis of complex traits.

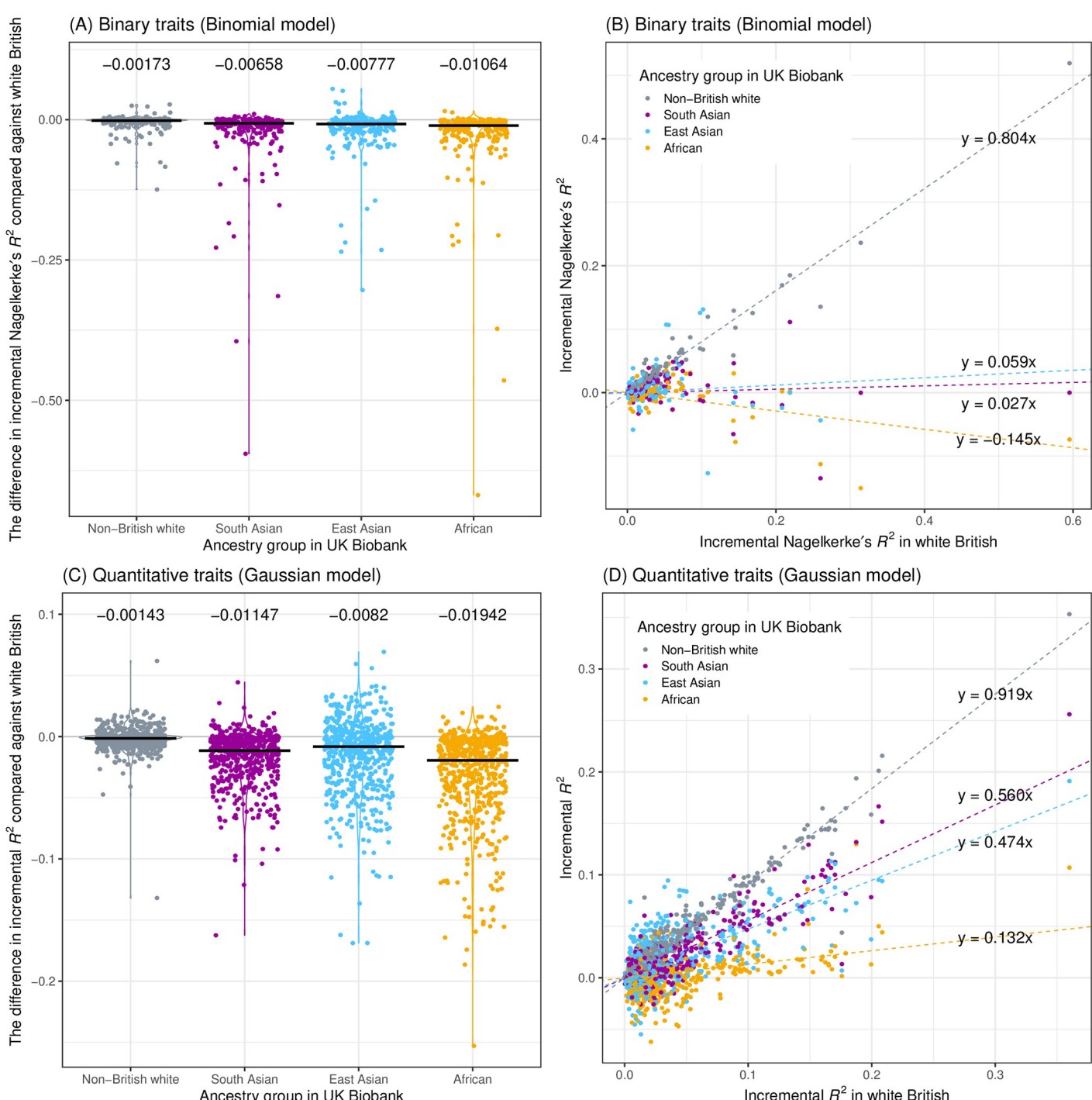

**Fig 6. Transferability assessment of PRS models across ancestry groups in the UK Biobank.** The incremental predictive performance (Nagelkerke's pseudo-$R^2$ for 244 binary traits (A, B) and incremental $R^2$ for 569 quantitative traits (C, D)) was quantified in individuals in different ancestry groups in the UK Biobank and was compared against that in the hold-out test set constructed from the individuals in white British ancestry group. (A, C) the difference in the incremental predictive performance between the target group (x-axis, double-coded with color) and the source white British cohort. The median values are shown as black horizontal bars and numbers. (B, D) comparison of the incremental predictive performance in the target group (color) and the test set. A linear regression fit was shown for each ancestry group with the dashed lines. The slopes of the regression lines were also shown.

## Methods

### Ethics statement

This research has been conducted using the UK Biobank Resource under Application Number 24983, "Generating effective therapeutic hypotheses from genomic and hospital linkage data" (http://www.ukbiobank.ac.uk/wp-content/uploads/2017/06/24983-Dr-Manuel-Rivas.pdf). Based on the information provided in Protocol 44532, the Stanford IRB has determined that the research does not involve human subjects as defined in 45 CFR 46.102(f) or 21 CFR 50.3 (g). All participants of the UK Biobank provided written informed consent (more information is available at https://www.ukbiobank.ac.uk/2018/02/gdpr/).

### Study population and genetic data

UK Biobank is a population-based cohort study collected from multiple sites across the United Kingdom [18]. To minimize the variabilities due to population structure in our dataset, we restricted our analyses to unrelated individuals based on the following four criteria [46,55] reported by the UK Biobank in sample QC file, "ukb_sqc_v2.txt": (1) used to compute principal components ("used_in_pca_calculation" column); (2) not marked as outliers for heterozygosity and missing rates ("het_missing_outliers" column); (3) do not show putative sex chromosome aneuploidy ("putative_sex_chromo- some_aneuploidy" column); and (4) have at most ten putative third-degree relatives ("excess_relatives" column).

Using a combination of genotype principal components (PCs), the self-reported ancestry (UK Biobank Field ID 21000, https://biobank.ndph.ox.ac.uk/ukb/field.cgi?id=21000), and "in_white_British_ancestry_subset" column in the sample QC file from UK Biobank, we subsequently focused on people of self-identified white British (n = 337,129), self-identified non-British white (n = 24,905), African (n = 6,497), South Asian (n = 7,831), and East Asian (n = 1,704) ancestry as described elsewhere [23]. Briefly, we used a two-step procedure to define the five groups. We first used the genotype principal component loadings of the individuals and set thresholds on component 1 and component 2 as follows: (1) self-identified White British: $-20 \leq PC1 \leq 40$ and $-25 \leq PC2 \leq 10$ and in_white_British_ancestry_subset == 1; (2) self-identified non-British White: $-20 \leq PC1 \leq 40$, $-25 \leq PC2 \leq 10$, has a self-reported ancestry of White, and does not identify themselves as White British; (3) African: $260 \leq PC1$, $50 \leq PC2$, and does not identify themselves as any of the following: Asian, White, Mixed, or Other population groups; (4) South Asian: $40 \leq PC1 \leq 120$, $-170 \leq PC2 \leq -80$, and does not identify themselves as any of the following: Black, White, Mixed, or Other population groups; and (5) East Asian: $130 \leq PC1 \leq 170$, $PC2 \leq -230$, and does not identify themselves as any of the following: Black, White, Mixed, or Other population groups. To refine the population definition by removing the outliers, we computed population-specific genotype PCs using approximately LD independent ($R^2 < 0.5$) common (population-specific minor allele frequency > 5%) biallelic variants outside of the major histocompatibility complex region [23]. We applied following thresholds [23]: (1) South Asian: $-0.02 \leq$ population-specific $PC1 \leq 0.03$, $-0.05 \leq$ population-specific $PC2 \leq 0.02$; and (2) East Asian: $-0.01 \leq$ population-specific $PC1 \leq 0.02$, $-0.02 \leq$ population-specific $PC2 \leq 0$.

We randomly split the white British cohort into 70% training (n = 235,991), 10% validation (to select the optimal sparsity level) (n = 33,713), and 20% test (n = 67,425) sets [23,56]. We used the same split of training, validation, and test set for all tested traits. The non-British white, African, South Asian, and East Asian samples were only used as test sets.

## Variant quality control and variant annotation

We used genotype datasets (release version 2 for the directly genotyped variants and the imputed HLA allelotype datasets) [19], the CNV dataset [22], and the hg19 human genome reference for the main PRS analyses in the study. Additionally, we considered imputed variants (release version 3) to investigate whether the imputed variants would improve the predictive performance. We annotated the directly-genotyped variants using Ensembl's Variant Effect Predictor (VEP) (version 101) [57,58] with the LOFTEE plugin (https://github.com/konradjk/loftee) [49], for which we created a Docker container image (https://github.com/yk-tanigawa/docker-ensembl-vep-loftee). Using ClinVar (version 20200914) [28], we annotated "pathogenic" and "likely pathogenic" variants.

We performed variant quality control as described elsewhere [23,46,55]. Briefly, we focused on the variants passing the following criteria: (1) outside of the major histocompatibility complex (MHC) region (hg19 chr6:25477797–36448354); (2) the missingness of the variant is less than 1%, considering that the two genotyping arrays (the UK BiLEVE Axiom array and the UK Biobank Axiom array) cover a slightly different set of variants [19]; (3) the minor-allele frequency is greater than 0.01%; (4) Hardy-Weinberg disequilibrium test p-value is less than $1.0 \times 10^{-7}$; (5) Passed the comparison of minor allele frequency with the gnomAD dataset (version 2.0.1) as described before [46,49]; (6) We manually investigated the cluster plots for a subset of variants and removed 11 variants that have unreliable genotype calls [46].

We grouped the VEP-predicted consequence of the variants into six groups: protein-truncating variants (PTVs), protein-altering variants (PAVs), proximal coding variants (PCVs), Intronic variants (Intronic), variants in the untranslated region (Intronic), and other variants (Others). Our grouping rule of the VEP-predicted consequence is summarized in (S4 Table).

We included the imputed copy number variants (CNVs) [22] and imputed HLA allelotypes [21]. The CNVs were called using PennCNV (v.1.0.4) [59] on raw signal intensity data from each genotyping array as described elsewhere [22]. Because the precise location of the CNVs is not identified, we did not infer the functional consequences of CNVs with variant annotation. The HLA allelotypes at HLA-A, -B, -C, -DPA1, -DPB1, -DQA1, -DQB1, -DRB1, -DRB3, -DRB4, and -DRB5 loci were imputed using the HLA*IMP:02 and imputed dosage file is provided by the UK Biobank. We included 156 alleles across all 11 loci that had a frequency of 0.1% or greater in the white British. We rounded allele dosage when they were within plus or minus 0.1 of 0, 1, or 2. We excluded the remaining nonzero entries. We also excluded erroneous total allele counts post-rounding [21].

When evaluating whether the inclusion of the imputed variants would improve the predictive performance of the PRS models, we focused on the 5,931,362 imputed variants [19] based on the following criteria: (1) imputation INFO score is greater than 0.7, (2) minor allele frequency computed across the entire ∼500k genotyped samples (UK Biobank Resource 1967, https://biobank.ctsu.ox.ac.uk/crystal/refer.cgi?id=1967) is greater than 0.01, (3) biallelic variants, (4) the variant is not present in the directly genotyped variants, and (5) missingness is less than 1%. We subsequently combined the imputed variant dataset with the directly genotyped variants, imputed HLA allotypes, and copy number variants.

## Phenotype definitions in the UK Biobank

We analyzed a wide variety of traits in the UK Biobank, including disease outcome [46,60], family history [46,60], cancer registry data [46], blood and urine biomarkers [23], hematological measurements, and other binary and quantitative phenotypes [55,56]. Some phenotype information collected at UK Biobank's assessment center contains up to four instances, each of which corresponds to (1) the initial assessment visit (2006–2010), (2) first repeat assessment

visit (2012–2013), and (3) imaging visit (2014-), and (4) first repeat imaging visit (2019-). Briefly, for binary traits, we performed manual curation of phenotypic definitions and assigned "case" status if the participants are classified as the case in at least one of their visits and "control" otherwise. For quantitative traits, we took the median of non-NA values, as described elsewhere [55].

Previously, we analyzed blood and urine biomarker traits, investigating the effects of covariates on the biomarker levels and derived covariate-adjusted biomarker values [23]. Briefly, we used a linear regression model to account for the covariate effects on the log-transformed measurement values from UK Biobank and adjusted for principal component loadings of genotype, age, sex, age by sex interactions, self-identified ancestry group, self-identified ancestry group by sex interactions, fasting time, estimated sample dilution factor, assessment center indicators, genotyping batch indicators, time of sampling during the day, the month of assessment, and day of the assay. We used the PRS models trained for the covariate-adjusted traits [23]. To quantify the incremental predictive performance against the covariate-only models, we quantified predictive performance against the original measurement values, except eGFR, AST/ALT ratio, and non-albumin protein, where we used the covariate-adjusted trait values. Those three traits are derived from covariate-adjusted biomarkers [23] and do not have raw measurement values.

The list of 1,565 traits with at least 100 cases (for binary traits) or non-NA measurements (for quantitative traits) analyzed in this study is listed in (S1 Table).

## Construction of sparse PRS models

Using the batch screening iterative Lasso (BASIL) algorithm implemented in the R *snpnet* package [10], we constructed the sparse PRS models for the 1,565 traits. We used the Gaussian family and the $R^2$ metric for quantitative traits, whereas we used the binomial family and the AUC-ROC metric for the binary traits [10]. For each trait, we fit a series of regression models with a varying degree of sparsity on the training set, consisting of 70% (n = 235,991) of unrelated individuals of white British ancestry. The predictive performance of each of the models is evaluated on the validation set, which consists of 10% (n = 33,713) of unrelated individuals of white British ancestry to guide the selection of the optional level of sparsity. We selected the sparsity that maximizes the predictive performance in the validation set. We subsequently refit the penalized regression model using the individuals in the combined training and validation set individuals (n = 269,704), which we denote as score development set, to maximize the power in the regression model [10]. We used the same training, validation, and test set split across all the PRS models analyzed in this study.

As opposed to many PRS methods that operate on the GWAS summary statistics [3–9,13–15], our method takes individual-level genotype and phenotype data. Using $L_1$ penalized regression (also known as Lasso), BASIL simultaneously performs variable selection and effect size estimation of the selected variants. We included age, sex, and top ten population-specific genotype PC loadings computed for the white British individuals [23] as unpenalized covariates. Thanks to the $L_1$ penalty term in the objective function that penalizes the number of features of non-zero regression coefficients, the resulting models will be sparse, meaning that they will have fewer genetic variants than unpenalized models [10].

To prioritize coding variants over non-coding variants in linkage, we assigned three levels of penalty factors (also known as penalty scaling parameter) [61]: 0.5 for pathogenic variants in ClinVar [52] or protein-truncating variants according to VEP-based variant annotation [58]; 0.75 for likely pathogenic variants in ClinVar, VEP-predicted protein-altering variants, or imputed allelotypes; and 1.0 for all other variants. The assignment rules of the penalty

factors are summarized in (S5 Table). The variants with lower values of penalty factors are prioritized in the $L_1$ penalized regression. To assess the degree of prioritization of the medically relevant alleles and their impacts on the predictive performance, we focused on four traits (standing height, BMI, high cholesterol, and asthma) and fit a separate model without penalty factors. We compared the number of selected variants and the predictive performance.

## Predictive performance and transferability of PRS models

We evaluated the predictive performance ($R^2$ for quantitative traits and Nagelkerke's pseudo-$R^2$ [also known as Cragg and Uhler's pseudo-$R^2$] [24,25] for binary traits) of PRS models (S6 Table). For the binary traits, we also evaluated the receiver operating characteristic area under the curve [ROC-AUC] and Tjur's Coefficient of Discrimination (Tjur's pseudo-$R^2$) [26]. For $R^2$ and ROC-AUC, we evaluated the 95% confidence interval of predictive performance using approximate standard error of $R^2$ [62,63] and DeLong's method [64], respectively. We used the individuals in the hold-out test set (n = 67,425) of white British ancestry as well as additional sets of individuals in non-British white (n = 24,905), African (n = 6,497), South Asian (n = 7,831), and East Asian (n = 1,704) ancestry groups. We evaluated the predictive performance of (1) the genotype-only model, (2) the covariate-only model, and (3) the full model that considers both covariates and genotypes. We computed the difference between the full model and the covariate-only model to derive the incremental predictive performance.

To evaluate the predictive performance of the covariate-only model in the hold-out test set of white British ancestry, we fit a generalized regression model, trait $\sim$ age + sex + array + Genotype PCs, using the individuals in the score development set. We subsequently computed the risk scores based on the covariate terms for the individuals in the hold-out test set. The array is an indicator variable denoting the types of the genotyping array (either the UK BiLEVE Axiom array or the UK Biobank Axiom array). For the individuals in non-British white, African, South Asian, and East Asian ancestry groups, we took the ancestry group-specific PCs computed for each set [23] and fit the same regression model for each group. We did not use the array indicator variable for African, South Asian, and East Asian because all individuals in those ancestry groups were genotyped on the UK Biobank Axiom Array (S2 Table).

To evaluate the predictive performance of the genotype-only model, we computed the polygenic risk score for the sets of individuals for evaluation using the--score command implemented in plink2 [65]. We quantified evaluation metrics ($R^2$, Nagelkerke's pseudo-$R^2$, ROC-AUC, and Tjur's pseudo-$R^2$).

To evaluate the predictive performance of the full model, we fit a model, trait $\sim$ 1 + covariate-only score + PRS, using the covariate-only score and PRS described above. The constant term accounts for the potential differences in the trait mean (for quantitative traits) or case prevalence (for binary traits) between the score development population and the target population. We looked at the p-value reported for the PRS term for the statistical significance of the PRS model. We used $p < 2.5 \times 10^{-5}$ (= 0.05/2000, adjusted for multiple hypothesis testing using the Bonferroni method for the number of traits analyzed in the study) as the significance threshold.

We also computed the difference in $R^2$ or Nagelkerke's pseudo-$R^2$ between the full and covariate-only models to derive the incremental predictive performance.

## SNP-based Heritability estimation

To compare the incremental predictive performance of the PRS models with SNP-based heritability, we applied genome-wide association analysis with PLINK. Specifically, we applied--glm command in PLINK [65] v2.00-alpha with age, sex, array, the number of CNVs, the length

of CNVs, and the top ten genotype PC loadings as covariates. The array is an indicator variable denoting whether the UK Biobank Axiom array or UK BiLEVE Axiom array was used in the genotyping. We included this term if the variants were directly measured on both arrays. The number and the length of the CNVs are described elsewhere [22]. The genotype PCs are the principal component (PC) loadings of individuals. We computed the population-specific PCs using the unrelated individuals in white British and used the first 10 PCs [23]. In the regression analysis, we standardized the variance of the covariates (--covar-variance-standardize option) and applied quantile normalization for the quantitative phenotype (--pheno-quantile-normalize option). Note, we did not perform quantile normalization in the PRS analysis. We used "cc-residualize" and "firth-residualize" options that implement the approximation [66] for efficient computation of GWAS p-values. We subsequently applied linkage disequilibrium (LD) score regression (LDSC) [27] and characterized the SNP-based heritability (S7 Table). We compared the predictive performance of the PRS models and the LDSC-based heritability estimates.

## Correlation analysis of the number of genetic variants and predictive performance of PRS models

We applied Spearman's correlation test implemented in R to assess the rank correlation between the size (the number of genetic variants included in the model) and the effect size (the incremental predictive performance) of the PRS model.

## Statistics

For computational and statistical analysis, we used Jupyter Notebook [67], R [68], R tidyverse package [69], and GNU parallel [70]. The p-values were computed from two-sided tests unless otherwise specified.

## Supporting information

**S1 Fig. Statistical significance of the assessment center terms in phenotype prediction.** We fit a regression model on age, sex, the types of genotyping arrays, polygenic risk score, and assessment centers for each of the 1,565 traits analyzed in the study. The frequency of the statistical significance ($-\log_{10}(P)$) of assessment center variables was shown. The cumulative frequency was shown on the secondary axis on the right. The statistical significance after the Bonferroni correction was shown as a red vertical line.
(TIF)

**S2 Fig. The impact of prioritizing the medically relevant alleles with penalty factors on the predictive performance of *snpnet* PRS models.** The predictive performance (AUC for binary traits and $R^2$ for quantitative traits) evaluated across hold-out test set individuals of different ancestry groups in UK Biobank are shown for four traits. The error bars represent the 95% confidence interval.
(TIF)

**S3 Fig. The impact of the imputed genetic variants on the predictive performance of *snpnet* PRS models.** The predictive performance (AUC for binary traits and $R^2$ for quantitative traits) evaluated across hold-out test set individuals of different ancestry groups in UK Biobank are shown for four traits. The error bars represent the 95% confidence interval.
(TIF)

**S1 Table. List of traits analyzed in the study and the predictive performance of the corresponding PRS models.** For the 1,565 traits analyzed in the study, the following information is shown: trait category, the phenotype ID in Global Biobank Engine (GBE ID), trait name, the types of link functions in a generalized linear model (Gaussian for quantitative traits and Binomial for binary traits), the predictive performance of the genotype-only model, covariate-only model, the full model that considers both genotype and covariates, as well as the incremental predictive performance (Delta[Full, covariates-only]), the number of genetic variants included in the PRS model, the statistical significance of the incremental predictive performance in a hold-out test set consists of a subset of white British individuals in the UK Biobank, whether the p-value is significant after multiple-hypothesis correction ($p < 2.5 \times 10^{-5}$), the score ID in polygenic score (PGS) catalog, the experimental factor ontology term ID of the mapped traits in PGS catalog, and the label of the mapped traits in PGS catalog.
(XLSX)

**S2 Table. The cohort characteristics.** For each ancestry group in UK Biobank, the number of individuals (n), age (mean and standard deviation [sd]), sex (percentage of individuals in male), the fraction of individuals genotyped on the UK Biobank Axiom Array. The statistics for the white British ancestry group were shown for the 70% training set, 10% validation set, and 20% test set.
(XLSX)

**S3 Table. The number of variants with non-zero BETAs is shown across four traits.** For each trait, we compared two models: without and with penalty factors to prioritize the medically relevant alleles.
(XLSX)

**S4 Table. The variant consequence grouping.** We grouped the Ensembl's variant effect predictor (VEP)-predicted consequence of the genetic variants into six groups (Consequence group): protein-truncating variants (PTVs), protein-altering variants (PAVs), protein-coding variants (PCVs), intronic variants (Intronic), variants in untranslated region (UTR), and other non-coding variants (Others). The links to the sequence ontology (SO) term detailing the definition of each of the predicted consequences are shown.
(XLSX)

**S5 Table. The penalty factor assignment rule.** We used the VEP-predicted consequence and ClinVar annotation to prioritize protein-truncating, protein-altering, and (likely) pathogenic variants by assigning lower penalty factor values. The penalty factor and the number of variants stratified by genetic variants (genotype or allelotype), predicted consequence, and ClinVar annotation is shown.
(XLSX)

**S6 Table. The predictive performance of PRS models.** For each trait (Trait category, GBE_ID, and Trait Name), we show the types of link functions in a generalized linear model (GLM family column, Gaussian for quantitative traits and binomial for binary traits), the population split (population), the types of the predictive model (model column), the types of evaluation metric ($R^2$ [R2], Nagelkerke's pseudo-$R^2$ [NagelkerkeR2], AUROC [AUC], or Tjur's Coefficient of Discrimination [TjurR2]), the value of the specified metric and its lower and upper bound of 95% confidence interval, and the statistical significance (p-value).
(XLSX)

**S7 Table. Estimated SNP-based heritability.** For each trait with a significant PRS model (trait, trait_name, and trait_category), we show the types of link functions in a generalized

linear model (family column, Gaussian for quantitative traits and binomial for binary traits), estimated SNP-based observed scale heritability with standard error (h2_obs and h2_obs_se), lambda GC (lambda_GC), mean chi-square statistic (mean_chi2), LD score regression intercept and its standard error (intercept and intercept_se), and the proportion of the inflation attributed to the LD score regression intercept, defined by (intercept -1)/(mean(chi-square)-1), and its standard error (ratio and ratio_se).
(XLSX)

## Acknowledgments

Some of the computing for this project was performed on the Sherlock cluster. We would like to thank Stanford University and the Stanford Research Computing Center for providing computational resources and support that contributed to these research results. The content is solely the responsibility of the authors and does not necessarily represent the official views of the funding agencies; funders had no role in study design, data collection and analysis, decision to publish, or preparation of the manuscript.

## Author Contributions

**Conceptualization:** Manuel A. Rivas.

**Data curation:** Yosuke Tanigawa, Guhan Venkataraman, Johanne Marie Justesen, Manuel A. Rivas.

**Formal analysis:** Yosuke Tanigawa, Robert Tibshirani, Trevor Hastie, Manuel A. Rivas.

**Funding acquisition:** Robert Tibshirani, Trevor Hastie, Manuel A. Rivas.

**Investigation:** Yosuke Tanigawa, Robert Tibshirani, Trevor Hastie, Manuel A. Rivas.

**Methodology:** Yosuke Tanigawa, Junyang Qian, Robert Tibshirani, Trevor Hastie, Manuel A. Rivas.

**Project administration:** Robert Tibshirani, Trevor Hastie, Manuel A. Rivas.

**Resources:** Robert Tibshirani, Trevor Hastie, Manuel A. Rivas.

**Software:** Yosuke Tanigawa, Junyang Qian, Ruilin Li.

**Supervision:** Robert Tibshirani, Trevor Hastie, Manuel A. Rivas.

**Validation:** Yosuke Tanigawa, Robert Tibshirani, Trevor Hastie, Manuel A. Rivas.

**Visualization:** Yosuke Tanigawa.

**Writing – original draft:** Yosuke Tanigawa, Manuel A. Rivas.

**Writing – review & editing:** Yosuke Tanigawa, Junyang Qian, Guhan Venkataraman, Johanne Marie Justesen, Ruilin Li, Robert Tibshirani, Trevor Hastie, Manuel A. Rivas.

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
