## [Decision Letter · Decision Letter 0]

12 Oct 2021

Dear Dr Tanigawa,

Thank you very much for submitting your Research Article entitled 'Significant Sparse Polygenic Risk Scores across 428 traits in UK Biobank' to PLOS Genetics.

The manuscript was fully evaluated at the editorial level and by independent peer reviewers. The reviewers appreciated the attention to an important topic but identified some concerns that we ask you address in a revised manuscript

We therefore ask you to modify the manuscript according to the review recommendations. Your revisions should address the specific points made by each reviewer.

[LINK]

Yours sincerely,

Samuli Ripatti

Associate Editor

PLOS Genetics

Scott Williams

Section Editor: Natural Variation

PLOS Genetics

Reviewer's Responses to Questions

**Comments to the Authors:**

Reviewer #1: In this paper, the authors have applied BASIL, a method previously developed by the authors, to 1600 traits in the UK Biobank. They have also provided a Global Biobank Engine which shows the predictive power of their PRS. However, given the current state of this paper, I cannot recommend this for publishing, and here are my reasons:

1. Most of the methods were “described elsewhere”, reading the current paper, it is not possible for readers to know how exactly the PRS were calculated. It is also unclear how the authors incorporate the HLA allelotype, CNV data and how the penalty factors were applied to the BASIL model.

2. In addition, the authors did not provide any details of how they obtained the GWAS summary statistics required for PRS calculation (presumably using the 70%UK biobank data using PLINK?)

3. Usually, we would include the UK Biobank assessment centre as a covariate to UK Biobank related analysis to avoid systematic collection error. In addition, for blood biomarkers, we usually want to include Fasting time, dilution factor and statin use, as those usually have significant impact to the model fit.

4. Was the metric reported based on the test sets?

5. How did the authors use the PCS and self-reported ancestry to identify the sample population? K mean clustering on PC1 and PC2? Or did they performed calculate the Euclidian distance between each sample and the PC centroid of each self-reported cluster?

6. Given the small sample size of non-European samples, does the author also split them into validation and test sets, or were the training all done on the validation sets? Based on page 10 line 230-242, I am guessing that the GWAS is performed on the British white, and parameter optimization / variable selections were done on the British white and then the predictive performance is performed on each of the populations? Or were the parameter optimization / variable selections also done in each of the populations?

7. Looking at the results shown in the Global Biobank Engine, there are many traits where the covariate has a predictive performance of 0 (assuming this is measured in R2). Specifically, for Lipoprotein A, its PRS performance is as high as 0.57 but the covariate has performance of 0, which is hard to believe.

8. The correlation of number of selected variable and the predictive performance sounds like an issue with power. This is similar to the self-contained test-statistics in gene set studies where including more information has a higher chance of having a high predictive performance. The lack of correlation in binary traits might be due to rare variants that have large effect or ascertainment of case control. For example, only one variants were selected for Iritis. Also, if we look at the Global Biobank Engine, we can see a lot of duplicated traits that were assigned to different categories. For example, Lipoprotein A is both a biomarker and blood assays. Were this duplicated removed from the correlation analyses? If not, then duplicated traits or highly correlated traits (e.g. Hand grip strength left and right) might have inflated the correlation. Considering the trait definition, it is also much easier to have duplication and correlation between quantitative traits than the binary triats.

Reviewer #2: Here the authors investigate properties of PRS across 428 traits in the UK. The work is nicely conducted and explained.

The introduction and Discussion need to better sign post what this study IS and what it is NOT about. For example, it is NOT promoting BASIL at the best PRS method, but rather it can be considered as a method that can be easily applied across many traits and could be useful across a range of genetic architectures without explicitly modelling genetic architectures. Also it is NOT proposing the best predictor for a trait because a) it only uses UKB data and not other GWAS data available for some traits b) relatives are excluded which (although independence of discovery and test sample are important) the GWAS discovery could be more powerful by including relatives (I am not saying you need to include the relatives for the purpose of this paper but rather more clearly define its boundaries). The purpose of this study is more about considering the properties of PRS of many traits from the same data set and examining trends across the traits.

1. Line 40 Define “sparse PRS”, this may be unclear to some readers.

2. Lines 55-61 do not define how you made discovery, tuning and testing samples, although the info is in the methods a brief summary is needed to interpret results presented

3. Figure 1 Axis labels too small – especially part D- lake plot – not an informative title; The entries of column 1 are not self-evident in terms of discovery/target, from lines 139-145 I see that other ancestries were not included in discovery sample, but not obvious from Fig 1 legend. Line 206, add to avoid ambiguity “The non-British white, African, Sout Asian and East Asian samples were only used as test sets”

4. I think “ethnic” is now regarded as cultural, and the preferred term in this context is “ancestry”

5. Figure 5 would benefit from “quotable” mean number stats for each ancestries

6. It would be of interest to have a plot of x-axis SNP-based heritability, y-axis increase in r2/AUC.

7. Given the differing sizes of test sets across ancestries it would be good to remind readers how this does/does not impact on interpretation of cross-ancestry comparisons.

Reviewer #3: In this paper Tanigawa et al. describe the systematic creation of polygenic risk scores (PRS) for > 1,600 traits using data from the UK Biobank (UKB). The construction and evaluation of the PRS is well-described, and the main result of the manuscript is a large resource of PRS built using a single method and a comprehensive web portal describing the results that will be useful for others looking to better understand the performance of each score and apply them to other cohorts. I do not have any major concerns about the manuscript; however, I think some of the unique features of the analysis should be better described and contextualised:

• The choice of variants (directly genotyped, imputed HLA, and CNVs) is quite different from classical PRS analyses that usually employs the full-set of imputed variants with MAF/INFO filtering. Does the performance improve if these imputed variants are included in the dataset? It is probably relevant to list the genotyping arrays employed, and adjust for the different arrays used in the performance evaluation.

• The prioritization of medically-relevant (ClinVar pathogenic/likely-pathogenic, VEP predicted protein-truncating/altering variants) for non-zero effect weights in the PRS is also a quite interesting addition; however, I was surprised to see no quantitative analysis of its impact on PRS performance. I would also hypothesize that the weighting would also impact the number of variants selected in the model (Figure 4)? Some comparison of the PRS performance and transferability with/without the variant prioritisation is necessary.

• Are there any obvious reasons that the correlation of predictiveness and number of variants changes? Is it dictated by differences in effect-size distributions or the MAF of selected variants?

Minor comments:

• A table with the age/sex/follow-up time/ancestry breakdown of the different training and test sets should be included. Were individuals included in the 70% training set consistent across all PRS being built?

• Description of how the p-value threshold for incremental predictiveness was selected should be provided.

• A major advantage of the BASIL/snpnet application in comparison to other PRS-derivation methods seems to be that it does not rely on LD reference panels which often limit the PRS derivation set to being a single-ancestry group. Given that the manuscript is somewhat focused on transferability of sparse PRS: would it be possible to derive new PRS using a random sample of the entire cohort (all ancestries) and evaluate how the multi-ancestry PRS compare to European-PRS at the whole population and single-ancestry level? [I realize this is beyond the scope of the current analysis but would be informative and may greatly improve the impact]

**Have all data underlying the figures and results presented in the manuscript been provided?**

Reviewer #1: Yes

Reviewer #2: Yes

Reviewer #3: Yes

PLOS authors have the option to publish the peer review history of their article (what does this mean?). If published, this will include your full peer review and any attached files.

Reviewer #1: **Yes: **Shing Wan Choi

Reviewer #2: No

Reviewer #3: No

---

## [Decision Letter · Decision Letter 1]

30 Nov 2021

Dear Dr Tanigawa,

Thank you very much for submitting your Research Article entitled 'Significant Sparse Polygenic Risk Scores across 813 traits in UK Biobank' to PLOS Genetics.

The manuscript was fully evaluated at the editorial level and by independent peer reviewers. The reviewers appreciated the attention to an important topic but identified some concerns that we ask you address in a revised manuscript

We therefore ask you to modify the manuscript according to the review recommendations. Your revisions should address the specific points made by each reviewer.

[LINK]

Yours sincerely,

Samuli Ripatti

Associate Editor

PLOS Genetics

Scott Williams

Section Editor: Natural Variation

PLOS Genetics

Reviewer's Responses to Questions

**Comments to the Authors:**

Reviewer #1: This manuscript has certainly been improved with the addition of more detail descriptions of the method and procedure involved. Thank you to the authors for making all these efforts.

Overall, I am still slightly confused as to what are the main messages of the current paper. I am also slightly concern about some interpretation of the results.

1. While the authors have now included much of the needed details regarding the procedure and methods performed, there are still some critical information that are missing. For example, quantitative traits were calculated as the “median of non-NA values, as describe elsewhere”, does that mean that the authors took the measurement across multiple assessment timepoint and take the median of that? Did the author perform any quality controls on the phenotype to remove outliers?

2. In a similar vein, for the blood and urine biomarkers, the covariate adjusted phenotype were calculated using the log transformed phenotypic value and the incremental predictive performance were calculated against the predictive value based on the original measurement. Were the original measurements also log transformed? Or was the untransformed value being used? If it is the latter, wouldn’t that introduce some bias? In addition, it is not uncommon to have blood or urine biomarker measurement of 0. In those scenarios, log transformation will lead to undefined value. How was that accounted for?

3. For the SNP-heritability estimates, the authors perform GWAS on the quantile normalized phenotype. Were the phenotypes also log transformed? It is difficult to assess the relationship between the PRS performance and SNP-heritability if they were performed on phenotypes undergone different transformation. Also, was the quantile normalization done on both quantitative traits and binary traits?

4. It is odd to have PRS that report a higher predictive performance than SNP-heritability, as the SNP-heritability are the theoretical upper bound of the PRS. It will be helpful if the authors can provide an explanation as to why the PRS performance is higher than the SNP-heritability (possibly due to different phenotypic transformation, or that the PRS include information that were excluded from the SNP-heritability estimate?). Standard error of the predictions should ideally be also reported to provide a better understanding of the power.

5. Based on how this paper is structured, it seems like the main message is that there is a significant positive correlation between the number of active variables in the PRS model and the incremental predictive performance in quantitative traits but not in binary traits, and this “highlighting the presence of diverse genetic architecture across disease outcomes.”. However, because the population prevalence of the binary traits is usually not known, and that the UK Biobank is a prospective cohort where the case numbers might not reflect the true population prevalence, the prediction performance of the binary traits, and their SNP-heritability estimations will likely be biased by ascertainment. In addition, in the main analysis, the authors “used the same split of training, validation and test set for all tested traits.”, which means that the case control ratio for the binary traits are likely different between the different set of samples, leading to a greater disparity of performance. Considering the lower heritability of binary traits (mean = 0.04 for binary trait, mean = 0.23 for quantitative traits, based on provided supplementary), reporting on observed instead of liability scale, and the different level of ascertainment bias, it is not surprising that the correlation between the number of active variables in the PRS model and the incremental predictive performance in binary traits are not significant. And it might be slightly misleading to conclude that the lack of correlation in binary traits, but in quantitative traits is a result of “the presence of diverse genetic architecture”.

6. Similar to the above comment, the case control ratio in different population might also differ, which was not accounted for here.

Other minor comments:

1. On line 197, line 249 and line 532, a different style of citation seems to be used? (ref:[#] , instead of [#])

2. For figure 4 top right, are the range inclusive or exclusive? E.g. for sample at 10 percentile, will they be grouped in [0-10%] or [10-20%]? Also, for multipaned plots, might be easier if the individual sub-plots are also labeled (e.g. 4a, 4b, 4c)

Reviewer #2: The authors have addressed my comments, but the revision has introduced some strong statements in the discussion which I believe are scale and power dependent. Therefore, I have additional comments.

New Figure 2A. For binary traits estimates of SNP-based heritability depends on proportion of GWAS discovery sample are cases, and Pseudo-R2 depend on the proportion of the target sample are cases. Although requiring a user-specified lifetime risk it would make more sense for these axes to be on the liability scale (even if lifetime risk used is the proportion of cases in the sample since all traits are in UKB) since then both axes are on the same scale and comparisons across traits are more valid.

Figure 5A and Figure 6 LHS use “incremental AUC”. AUC has the nice property that it doesn’t depend on the proportion of cases in the sample, both other than that it has very non-linear properties with respect to quantitative genetic metrics of polygenic traits such as heritability. For example, while a linear relationship might be expected in incremental R2 for quantitative traits (Figure 6 bottom left quadrant) I wouldn’t expect a linear relationship in incremental AUC. This may impact the conclusion line 331 “we found a significant correlation across quantitative traits but not within binary traits” Suggest of these analyses R2 liability is used.

The point being made here “While the underlying genetic architecture of binary traits may span the gamut of a wide variety of polygenicity, that of highly heritable quantitative traits may not be compatible with monogenic inheritance as illustrated in the wide adoption of Fisher’s infinitesimal model”. That is a very broad statement not really relevant to the study, suggest delete. Moreover, expressions of genes are quantitative traits that likely span the gamut of genetic architectures.

I am concerned about the new conclusions that contrast binary traits with quantitative traits with only a nod to differences in power. It is intuitive that for the same N (ie UKB sample size) as the proportion of cases tends to zero the power of the sample for detection of association is reduced. I think Yang et al (2009) equation 3 could help quantify expectations doi:10.1002/gepi.20456

Supp Table 6 seems to have a column missing -across the labels in column A-D there are 3 sets of results. Model column? I have never seen TjurR2 presented before in this context. It is presented together with NagelkerkeR2. There is not justification as to why TjurR2 should be presented. Both I believe are dependent on the proportion of cases in the sample . Some of the AUC values seem implausibly high given the R2? Check?

Reviewer #3: The additional analyses and explanations in this revision result in a much improved manuscript describing the phenome-wide application of BASIL to derive PGS in UKB. The authors have addressed all my concerns (especially with respect to the description of variant-penalties), the analyses are technically sound and well described.

**Have all data underlying the figures and results presented in the manuscript been provided?**

Reviewer #1: Yes

Reviewer #2: Yes

Reviewer #3: Yes

PLOS authors have the option to publish the peer review history of their article (what does this mean?). If published, this will include your full peer review and any attached files.

Reviewer #1: **Yes: **Shing Wan Choi

Reviewer #2: No

Reviewer #3: No

---

## [Decision Letter · Decision Letter 2]

15 Feb 2022

Dear Dr Tanigawa,

We are pleased to inform you that your manuscript entitled "Significant Sparse Polygenic Risk Scores across 813 traits in UK Biobank" has been editorially accepted for publication in PLOS Genetics. Congratulations!

Yours sincerely,

Samuli Ripatti

Associate Editor

PLOS Genetics

Scott Williams

Section Editor: Human Variation

PLOS Genetics

Comments from the reviewers (if applicable):

Please address the one remaining request from the reviewer.

Reviewer's Responses to Questions

**Comments to the Authors:**

Reviewer #1: With the latest update, the authors have address most of my concerns. Thank you for the hard works.

Reviewer #2: Thank you for addressing the comments.

I understand your choices to report SNP-based heritability on the observed scale and Nagelkerke's R2. Please add a sentence to remind readers that both these metrics depend on the proportion of cases in the samples (discovery and target respectively) including in Figure legends.

**Have all data underlying the figures and results presented in the manuscript been provided?**

Reviewer #1: Yes

Reviewer #2: Yes

PLOS authors have the option to publish the peer review history of their article (what does this mean?). If published, this will include your full peer review and any attached files.

Reviewer #1: **Yes: **Shing Wan Choi

Reviewer #2: No

**Data Deposition**

http://datadryad.org/submit?journalID=pgenetics&manu=PGENETICS-D-21-01210R2

**Press Queries**

---

## [Editor Report · Acceptance letter]

28 Feb 2022

PGENETICS-D-21-01210R2 

Significant Sparse Polygenic Risk Scores across 813 traits in UK Biobank 

Dear Dr Tanigawa, 

We are pleased to inform you that your manuscript entitled "Significant Sparse Polygenic Risk Scores across 813 traits in UK Biobank" has been formally accepted for publication in PLOS Genetics! Your manuscript is now with our production department and you will be notified of the publication date in due course.

With kind regards,

Zsofia Freund

PLOS Genetics

On behalf of:
